# Before and Amid COVID-19 Pandemic, Self-Perception of Digital Skills in Saudi Arabia Higher Education: A Longitudinal Study

**DOI:** 10.3390/ijerph19169886

**Published:** 2022-08-11

**Authors:** Mostafa Aboulnour Salem, Wafaa Hassanien Alsyed, Ibrahim A. Elshaer

**Affiliations:** 1Deanship of Development and Quality Assurance, King Faisal University, Al-Ahsa 31982, Saudi Arabia; 2Deanship of Faculty Affairs, King Faisal University, Al-Ahsa 31982, Saudi Arabia; 3Department of Management, Faculty of Business Administration, King Faisal University, Al-Ahsa 31982, Saudi Arabia; 4Hotel Studies Department, Faculty of Tourism and Hotels, Suez Canal University, Ismailia 41522, Egypt

**Keywords:** digital skills (D.S.), self-perception, COVID-19, higher education, sustainable development goals (SDGs), 21st-century skills

## Abstract

Compatible with global sustainable development report, 2016 edition, and vision 2030, Saudi Arabia recognized the importance of technology in achieving the Sustainable Development Goals (SDGs). This paper aims to measure the self-perception of digital skills among students in Saudi Arabia’s higher education system to understand how they were influenced before and amid the COVID-19 pandemic. In 2019 before the COVID-19 pandemic, we started a project to study the self-perception of digital skills among Saudi Arabia university students (group A). A total of 469 students participated in this research. The validity and reliability of the employed scale were tested with first-order confirmatory factor analysis (CFA). The differences between the two groups (before and amid the pandemic) were tested through the Mann–Whitney U test. The results for group A (N = 232 students) showed a higher self-perception of their digital skills. In March 2020, amid the pandemic, Saudi Arabia closed and shifted to technology-based teaching like many other countries worldwide. After students’ return to universities in 2021, an evaluation of how the students perceived their own digital skills was again conducted (group B). The results for group B (N = 237 students) demonstrated a lower level of confidence in their own digital abilities. Comparing two groups (A and B), after the educational course was administered, group A (prior to COVID-19) had greater self-perceptions of digital skills than group B (amid COVID-19). Students’ perceptions of their own digital skills have been negatively impacted as a result of the pandemic situation caused by COVID-19. The collected evidence suggests that there is a difference, and that this difference is statistically significant. As a result of the substantial relationship between self-perception of digital skills and how students deal with reality based on their own self-perception, Saudi Arabia higher education ministry shifted teaching methods to be based on technology. Other significant findings and their implications for practice and theory were reported in this study. Finally, limitations and prospects for future research were also elaborated.

## 1. Introduction

In a fast-changing knowledge society, economic innovation starts with people, making human capital a powerful tool within a community and its organizations. The 21st-century digital skills, so-called “human skills”, drive societies’ competitiveness and innovation capacity [1]. One of the eight primary skills for lifelong learning identified by the European Union (EU) to successfully meet the challenges of developing a knowledge society and competing in the world market is digital skills (DS) [2]. Therefore, higher education alumni should own digital skills and be familiar with current open information and communication technologies (ICTs) tools; to meet labor market needs [3].

Saudi Arabia’s 2030 vision recognizes the technology dimension of the Sustainable Development Goals (SDGs) and technology’s importance in achieving the SDGs [4]. Technology is reflected in SDG17 as a key “means of implementation”. Among the 169 SDG targets, 14 explicitly refer to “technology”, and another 34 are related to issues most often discussed in terms of technology. There are also specific technology dimensions to the other remaining 121 targets. Technological progress has solved many ills and problems [5]. Furthermore, digital skills comprise around 70% of all fast-growing skills: data, digital, social, search, media, creative, and others. Today’s job market has shifted toward hybrid roles requiring multifaceted employees with a powerful digital grounding [6].

Until 2019, like higher education students worldwide, students in Saudi Arabia saw themselves as digitally oriented. They had a high self-perception of using ICT applications and entering digital environments. They had a high level of readiness to support achieving ICT with National Transformation Programs for Saudi Vision 2030 and the technology dimension of the (SDGs) [7,8,9,10].

Due to exceptional situations, in March 2020, teaching in Saudi Arabia shifted from traditional classroom teaching to hybrid learning, as in many other countries worldwide [11,12]. The COVID-19 pandemic caused universities to change their educational approach and embrace technology and virtual learning platforms. Many different types of research have been conducted worldwide showing the incidence of changes in attitudes, behaviors, and self-perceptions about digital skills, behaviors, enablement, and competencies among students [13,14,15]. The majority of the research looked at how students perceived their own digital skills either before the pandemic [11,12] or while it was in progress [13,14,15,16]. Nevertheless, there was not a single study that compared the situation before and during the pandemic. The current research filled this gap with a longitudinal study to test the self-perception of digital skills in Saudi Arabia’s higher education before and amid the COVID-19 pandemic. The results can assist in developing plans that could help in enhancing students’ digital skills.

## 2. Literature Review

### 2.1. Student Digital Skills

The International Telecommunication Union (ITU) [17] and the United Nations (UN) [5] introduced a definition of digital skills as “the ability to use information and communication technologies (ICTs) in ways that help individuals to achieve beneficial, high-quality outcomes in everyday life, for themselves and others, that reduce the potential harm associated with more negative aspects of digital engagement”.

The commonly recognized definition of DS highlights that digital skills are essential for the students’ ability to use ICTs and other digital media whose objective is the intelligent, adaptable, and secure enhancement of online and offline activities [18]. Knowledge of digital technologies is one of the primary forces behind the ongoing fourth industrial revolution. Augmented reality (AR), artificial intelligence (AI), big data analytics (BDA), cybersecurity, system integration, the internet of things, and autonomous robots are some of the identified enabling technologies, as stated in the national policy on industry 4.0 (National Transformation Program 2020) [3]; on the other hand, SDG 4 identifies digital skills as a target that seeks to increase the number of people with employment-relevant skills [19].

In the past, graduates possessed skills that were insufficient to compete in today’s updated knowledge economy [1]. Hard and soft skills are now required for entry into the workforce, with the latter also referred to as “21st-century skills,” which include the following: problem-solving communication skills, critical thinking abilities, creativity and innovation, collaboration with others, and digital skills [20].

Hence, the current article is focused on digital skills (DS) as part of 21st-century skills. Previous publications defined six sub-skills of the (DS): digital information skills, digital communication skills, digital collaboration skills, critical thinking digital skills, creative digital skills, and problem-solving digital skills, as in the following concepts (Table 1) [13,14,15,20,21,22,23].

Recent research indicates that DS are required to access the internet, as well as to analyze and utilize online content [20,21,22,23]. Moreover, research has revealed that some students have the digital skills to be digital natives while others are like digital immigrants [21,24,25,26]. Others indicate that most students learn and develop their digital skills outside of formal studies, which further supports the notion of digital skills being developed for other purposes than learning [20,27,28]. Comparatively, the context of the reviewed literature reveals that students of today expect to be able to use technologies for any purpose, including learning; they do not associate digital technologies or the internet with age, gender, self-perception, or level of education [27,28,29].

In the era of COVID-19, students’ informal digital technology skills are comparable to those required for learning; numerous studies have demonstrated a significant relationship or correlation between digital skills and online education [21,23,29].

### 2.2. Impact of COVID on Education in Saudi Arabia Universities

Until 2019, multiple studies and reports confirmed that Saudi Arabia undergraduate students perceive themselves as digitally oriented and prepared to use ICT applications and enter digital environments. In addition, they are highly prepared to support the ICT National Transformation Program for Saudi Vision 2030 and the technology dimension of the Sustainable Development Goals [30,31,32,33]. Due to exceptional situations, in March 2020, teaching in Saudi Arabia shifted from classroom teaching to hybrid learning, as in many other countries worldwide. Higher education has seen a radical shift since COVID-19’s inception, breaking apart many traditional approaches, such as the unity of space-time and the unity of action [12].

Because of the COVID-19 pandemic, universities underwent a paradigm shift in their approach to teaching and learning, recognizing the value of utilizing technology and virtual teaching platforms [34]. Moreover, multiple kinds of literature worldwide have shown incidence shifts in self-perception of the digital skills of higher education students [13,35,36,37].

## 3. Materials and Methods

### 3.1. Objectives

The researchers’ objective is to conduct a longitudinal study of students’ self-perception of digital skills in Saudi Arabia’s higher education system and the influence of the COVID-19 pandemic. An instrument has been developed to measure the level of the self-perception of digital skills among higher education students, and its reliability and validity assessed.

### 3.2. Sample

All students enrolled in the first year of a bachelor’s degree program at seven Saudi Arabian universities (King Faisal University (KFU), King Saud University (KSU), Qassim University (QU), King Khalid University (KKU), Northern Border University (NBU), Gazan University (GU), and Majmaa’ University (MU)) constituted the study’s sample, covering the main areas in Saudi Arabia, during two academic years 2019/2020 and 2020/2021. The students followed the Introduction to Computing Course (ICC) in the first semester. The sample consisted of 469 students: 232 students from the 2019–2020 academic year (group A) and 237 students from the 2020–2021 academic year (group B). Group A was the one that studied ICC before the pandemic, and group B studied ICC amid the pandemic. Incidental or convenience selection criteria were used based on students’ readiness to complete the questionnaire. It should be noted that both groups have taken the same course and that pre-and post-tests were administered to students. (See Table 2).

### 3.3. Instrument

An extensive literature review was conducted to obtain the study scale. The process yielded 30 items with five dimensions measuring the student’s digital skills. The scale showed good reliability and was derived from Rosenberg (1965) [38], Van Deursen and Van Dijk (2014) [39], Van Laar et al. (2019) [40]). Each competency was represented by a single item, and the most general concept encompassing all the specific digital skills content was selected. The 30 items that comprised the questionnaire for the study represent six dimensions: digital information skills (4-items, *a* = 0.812), digital communication skills (3-items, *a* = 0.804), collaboration digital skills (4-items, *a* = 0.811), critical thinking digital skills (3-items, *a* = 0.726), creative digital skills (3-items, *a* = 0.729), and problem-solving digital skills (5-items, *a* = 0.832). Each variable was operationalized on a 5-interval Likert scale, with students selecting one of five options to indicate the degree to which they reflect the stated perception. The instrument includes two sections. In the first section, the sociodemographic data of the respondents and several questions about their habits and activities were obtained. The organization of the second section reflects the general progression logic of digital skills levels. Six advanced digital skills were assessed to determine students’ digital skills: information digital skills, communication digital skills, collaboration digital skills, critical thinking digital skills, creative digital skills, and problem-solving digital skills [40]. The survey URL (https://2u.pw/xHJzo, accessed on 17 July 2022) (in English and Arabic) was circulated via participants’ private e-mails and social media in March 2022 and maintained for four weeks. Day-to-day, the investigators reviewed and observed the responses. Cronbach’s alpha coefficient and McDonald’s Omega were employed to assess the scale’s reliability, as shown in Table 3. CFA was conducted to determine the scale’s convergent and discriminant validity, as shown in Table 4.

### 3.4. Data Analysis Methods

Descriptive analysis was carried out to illustrate the respondents’ characteristics. The bivariate correlational analysis method was employed through Spearman’s Rho correlation coefficient to explain the correlations between the research variables. First order CFA tested scale validity and reliability. Furthermore, a contrast analysis was conducted to compare the mean scores collected in the first group (group A, before the pandemic) and the second group (group B, amid the pandemic). The non-parametric Mann–Whitney U method was employed with SPSS v.23 (SPSS Inc., Chicago, IL, USA).

## 4. Results

### 4.1. Descriptive Analysis Results

The majority of the students in both groups (group A and group B) were female; group A was 89.6% female, while group B was 89.4% female. The majority of respondents were between the ages of 18 and 20 (95.8% in group A and 98.4% in group B). A = 10.7% and B = 11.7% of students had never used technology as an educational tool.

Group A (prior to the pandemic) reported a usage rate of 42–73%, whereas group B reported a usage rate of 78–100%. In addition, when questioned about technology usage, group A reported use between 62% and 81% of the time, while group B reporteduse between 80% and 100% of the time. It is possible that the lockdown and virtual nature of classes may have contributed to this disparity. In both groups, a large proportion of the sample (group A = 91.1%; group B = 87.4%) was interested in becoming proficient in using various new software programmes and online resources.

### 4.2. Scale Validity and Reliability Results

High Alpha and Omega values can be defined as those between 0.8 and 1 (O’Dwyer, and Parker (2014)) [41]. A value of at least 0.7 is considered sufficient to ensure the instrument’s reliability. Table 3 shows that all alpha and omega values are greater than or equal to 0.7, indicating that the scale used is reliable.

To examine the convergent and discriminant validity of the study scale, a first-order confirmatory factor analysis (FCFA) test using Maximum Likelihood (ML) was conducted. Six dimensions of digital skills (information DS, communication DS, collaboration DS, critical thinking DS, creative DS and problem-solving DS) were subjected to CFA with their associated measuring items. Numerous goodness-of-fit (GoF) criteria were utilized to evaluate model fit as proposed by Kline (2015) [42]. The model has a good fit for the data: χ^2^ (204, N = 469) = 782.544, *p* < 0.001 (normed χ^2^ = 3.836, SRMR = 0.043, RMSEA = 0.038, CFI = 0.950, NFI = 0.952, and TLI = 0.959, PCFI = 0.850, and PNFI = 0.826). Composite reliability was used to evaluate construct reliability (CR). The study’s six dimensions have the following CR values, as shown in Table 4: information DS (0.863), communication DS (0.831), collaboration DS (0.811), critical thinking DS (0.785), creative DS (0.772) and problem-solving DS (0.889). All of the results were greater than the 0.70 thresholds, indicating a high level of internal consistency (Hair et al., 2017) [43].

Furthermore, the scores of the “average variance extracted” (AVE) for all six factors (information DS, communication DS, collaboration DS, critical thinking DS, creative DS and problem-solving DS) were 0.579, 0.558, 0.657, 0.719, 0.669, and 0.668 in that order (Table 4). All AVE values exceeded 0.50, indicating adequate convergent validity [43]. The values of AVE were also found to be greater than all “maximum shared variance” (MSV) scores (Table 4), indicating that the scale discriminant validity was sufficient [42].

### 4.3. Group Comparison Results

Table 5 compares the mean and standard deviation values of students’ six digital skills and self-perception before and after taking the Introduction to Computing Course (ICC) before and during the COVID-19 pandemic in two groups (A and B).

The results showed that group A (prior to COVID-19) students had greater self-perceptions of DS than group B (amid COVID-19). Following the educational intervention, there was a trend toward self-perceptions becoming more comparable, despite group A’s continuing to have more positive ones.

More than two samples were used in the Mann–Whitney U contrast tests to determine whether the differences in digital skills self-perceptions between groups A and B were statistically significant (Table 6).

According to the data collected, students’ pre-test digital skills self-perceptions were negatively impacted by the pandemic situation caused by COVID-19. It is possible to conclude with 95% certainty that there were statistically significant differences between groups A (prior to COVID-19) and group B (amid the COVID-19 pandemic).

As observed in Table 7, the changes always reflected an increase in the DS self-perception of students who completed the course prior to COVID-19 (Group A).

Similarly, we checked to see if there were significant differences between the two groups for the post-test in terms of self-perceptions (Table 8). Since almost all of the digital skill areas had a level significantly lower than 0.05, we could say with 95% certainty that the post-test self-perception of digital skills has changed significantly between before and during COVID-19.

When analyzing perceptions in the pre-test, the ranges showed that group A improved (Table 9), but there was a tendency to equalize after learning that could be related to the teaching obtained and technology usage after completing the ICT course.

Finally, a correlational analysis was conducted to test if there is a correlation between the time spent using technology and students’ DS self-perception in each dimension and the total perception of DS. When comparing groups, A and B, the Spearman Rho correlation coefficient was used in order to see how the two groups differed before and during the pandemic. Table 10 shows the results.

There was no correlation between usage time of ICT and all dimensions of DS self-perception in group A (prior to the pandemic). Group B, on the other hand, has a 99% confidence level that there is a positive statistical correlation. Nevertheless, the low strength could be justified by the crisis situation and the evidence that this group used ICT more as a learning tool, which positively affected their evaluation of their perception of DS.

In addition, the relationship between self-perceptions of digital skills and gender (male/female) and teaching experience was investigated. Regarding the gender variable, the data revealed a positive correlation between self-perceptions in the female section; however, the sample is predominantly female, so we decided to ignore any implications from this result. The results showed no clear statistical evidence of a correlation between the two variables, teaching experience and digital skills self-perception.

## 5. Conclusions

Lockdowns have had a significant impact on the digital transformation of universities, either because ICT expectations were very high or due to the fact that the delivery of ICT courses had never been introduced from an authentic theoretical educational base that extracts the real potentials of ICT. A global pandemic revealed its best side, highlighting the gap between many institutions’ development and strategic planning. If anything has been learned, it is that faculty and students need digital training.

Students are vulnerable to external problems that can affect their mental skills. This study shows how the transition to pandemic conditions affected self-perceptions of digital skills, which could be related to cognitive skills problems, stress, or anxiety. The previous argument was supported by Portillo et al. (2020) [44], Ma et al. (2021) [45], and Calderón, Ortí and Kuric (2022) [13], who found that the COVID-19 pandemic had a negative effect on academic aspects that influence lifestyles.

Before receiving the educational course (ICC), the non-pandemic group in this study showed a higher level of digital skills self-perception than the pandemic group. In a post-test, the group not in a crisis situation had more positive perceptions than the one affected by the COVID-19 crisis. These results are consistent with the results of Black (2021) [46], and Morina (2021) [47], who argued that when the uncontrolled external aspects shape a person’s self-perception, each person may retreat into their own inner world.

In this circumstance, it is evident that a global situation of exceptional magnitude may be the reason for the inequity between the two investigated groups. This inequity can be explained due to the strong relationship between student self-perception and how they recognize the world reality through their subjective vision and experiences [44,48]. It is well established that there is a direct link between learning and an individual’s self-perception of their level of competence as well as how that individual feels about themselves [45,49]. Therefore, factors such as discomfort, worry, and anxiety make learning difficult because they prevent students from correctly processing information and from perceiving all of the nuances of the knowledge, which ultimately has a negative impact on the students’ academic performance [46].

Simultaneously, the study findings revealed a positive statistical and significant correlation between the increased amount of time spent using technology (due to the pandemic) and a higher students’ self-perception of digital skills. The group that was formed before COVID can be differentiated from the group that was formed during COVID because of the obvious need for increased use of ICT as an educational method and because it is the only choice for students to maintain the education process. This results are in accordance with García-Vandewalle et al. (2021) [49], and Zhao, Llorente and Gómez. (2021) [50], who highlighted that self-perception is affected by four key factors: experiences or accomplishments of mastery, verbal persuasion, vicarious experiences, and physiological state. This explains the correlation between the increased amount of time spent using technology (due to the pandemic) and higher students’ self-perception of digital skills.

Furthermore, the results confirmed the direct relationship between the students’ self-perception of their digital skills and the pandemic. However, it cannot be assumed that gender is a factor in this case due to the lack of male representation in ICT education classrooms. It cannot even be considered an accurate representation of reality at this educational level. Furthermore, due to being forced to miss class, students use technology more frequently, influencing their perception of their digital skills after learning [29,44].

The current study showed evidence that the impacts of the pandemic expanded beyond the academic aspects such as class grades. The consequences of social distancing, self-isolation and fear of catching the virus are shifting our perception of ourselves and the surrounding environment, which may lead to negative learning consequences. The education system has a vital role as the base for gaining basic digital skills; therefore, it is essential to reorganize the study strategies in order to achieve a superior adaptation to the Saudi Vision 2030’s demands in digital skills [4,21]. We have to take advantage of the situation and go one step further, without conceiving the use of technology in teaching/learning during emergencies and crises—such as COVID-19—as a temporary and due to unforeseen circumstances. Rather, we should consider that everything that happened offers the foundations to build teaching strategies in which digital skills are considered equitably with other competencies, and focus on improved platforms, IT resources and other teaching tools. An improved model with diverse teaching formats, adequate instruction and resources that aid in skill acquisition is needed; in other words, a model reflecting the participation of the whole university in an actual digital transformation process is required. This model should be more flexible, open, and efficient [51,52].

This article provides evidence for the significant implications of self-perception of digital skills among students in Saudi Arabia’s higher education system before and amid the COVID-19 pandemic. The results may reflect the increased use of ICT as the only choice for students to maintain the education process. Therefore, the article recommends an additional review by policymakers, practitioners, digital learning providers, and investigators looking to develop efficient strategies concerning 21st-century skills, mainly digital skills, for higher education students in Saudi Arabia and Arab countries.

As a reminder of our own vulnerability and the unpredictable nature of the future, the year 2020 will go down in history as a historic, unforgettable year. This new reality revealed new responsibilities for the process of education and learning. Reflection on ourselves and our digital skills, as well as a renewed sense of life’s value, may lead us to a new understanding of the 21st century that values life itself, as well as our ability to connect with others.

Lastly, it is crucial to note that universities must plan strategies to alleviate the impacts of the pandemic in order to improve the digitalization skills of their students, thereby assisting them in solving their fundamental problems and assuming partial responsibility for their social impact. In this context, the findings point to a rise in digitalization skills among university students. However, these results may not be valid for broader populations. Therefore, future research can investigate self-perception of digital skills in different contexts and compare the results with the results of the current study.

## Figures and Tables

**Table 1 ijerph-19-09886-t001:** The study concepts and description.

Concepts	Description
Digital information skills	The ability to investigate, assess, and manage knowledge/information usually collected from multiple sources.
Digital communication skills	The capacity to communicate knowledge and ensure data is effectively described by understanding the digital environment and audience.
Collaboration digital skills	The ability to cooperate to transfer digital experiences, split tasks, and decode issues that could not be handled individually.
Critical thinking digital skills	The capability of thinking critically to judge and assess well the knowledge/information that is relevant for a given situation.
Creative digital skills	The provision of creative ideas to enable digital and emerging technologies through utilisation or development.
Problem-solving digital skills	Digital problem-solving skills for issues that are complex, uncertain, nonrecurrent, and require or benefit from digital solutions.

**Table 2 ijerph-19-09886-t002:** Sample of the Study.

Items	Frequency	Percentage
Gender		
Female	420	89%
Male	49	11%
**University**		
KFU	149	32%
KSU	68	14%
QU	52	11%
KKU	33	7%
NBU	88	19%
GU	51	11%
MU	28	6%
SUM	469	100%

**Table 3 ijerph-19-09886-t003:** Reliability coefficients.

Factors	A	Ω
Information DS	0.812	0.831
Communication DS	0.804	0.815
Collaboration DS	0.811	0.812
Critical thinking DS	0.726	0.753
Creative DS	0.729	0.773
Problem-solving DS	0.832	0.855
TOTAL	0.918	0.908

DS: Digital Skills.

**Table 4 ijerph-19-09886-t004:** Psychometric properties of the employed scale.

Factors	CR	Reference Value	AVE	Reference Value	MSV	Reference Value
Information DS	0.862	CR > 0.7	0.579	AVE > 0.5	0.421	MSV < AVE
Communication DS	0.831		0.558		0.345	
Collaboration DS	0.811		0.675		0.373	
Critical thinking DS	0.785		0.719		0.424	
Creative DS	0.772		0.669		0.430	
Problem-solving DS	0.889		0.688		0.437	

χ^2^ (203, N = 288) = 271.31, *p* < 0.001 (Normed χ^2^ = 1.336, SRMR = 0.043, RMSEA = 0.038, CFI = 0.950, NFI = 0.952, and TLI = 0.959, PCFI = 0.850, and PNFI = 0.826).

**Table 5 ijerph-19-09886-t005:** Mean and standard deviation for both groups.

Digital Skills	No_COVID Group		COVID Group		NO_COVID Group		COVID Group	
	Pre-Test		Pre-Test		Pos-Test		Post-Test	
	M	SD	M	SD	M	SD	M	SD
DS_1. Information	2.11	0.59	1.91	0.61	1.93	0.61	1.89	0.60
DS_2. Communication	2.51	0.63	2.21	0.63	2.19	0.69	2.23	0.69
DS_3. Collaboration	2.77	0.69	2.55	0.63	2.51	0.81	2.47	0.65
DS_4. Critical thinking	2.47	0.79	2.22	0.76	2.19	0.83	2.25	0.80
DS_5. Creative	2.93	0.81	2.61	0.83	2.49	0.92	2.57	0.83
DS_6. Problem-solving	2.51	0.81	2.51	0.74	2.06	0.93	2.36	0.79
General	2.55	0.59	2.38	0.55	2.21	0.69	2.30	0.60

M: Mean; SD: Standard Deviation.

**Table 6 ijerph-19-09886-t006:** Mann–Whitney U test with no-COVID-19/COVID-19 variable (pre-test).

Mann–Whitney U Test							
	DS_1	DS_2	DS_3	DS_4	DS_5	DS_6	General
U de Mann–Whitney	10,520.000	10,246.000	9917.000	11,683.000	10,368.500	13,365.000	10,318.000
W de Wilcoxon	24,716.000	24,442.000	24,063.000	25,779.000	24,364.500	27,561.000	24,514.000
Z	−3.464	−3.800	−4.100	−1.856	−3.642	−0.108	−3.673
Sig. Asymptotic (bilateral)	0.001	0.000	0.000	0.063	0.000	0.005	0.000

Grouping item: no COVID-19/COVID-19; DS_1. Information DS; DS_2. Communication DS; DS_3. Collaboration DS; DS_4. Critical thinking ds; DS_5. Creative ds; DS_6. Problem-solving DS.

**Table 7 ijerph-19-09886-t007:** Pre-test average ranges by DS factors.

Factors	COVID-19/No-COVID-19	Average Range
DS_1. Information DS	COVID-19	181.47
	non-COVID-19	143.52
DS_2. Communication DS	COVID-19	186.19
	non-COVID-19	146.89
DS_3. Collaboration DS	COVID-19	183.88
	non-COVID-19	141.29
DS_4. Critical thinking DS	COVID-19	181.89
	non-COVID-19	145.64
DS_5. Creative DS	COVID-19	184.42
	non-COVID-19	146.62
DS_6. Problem-solving DS	COVID-19	173.57
	non-COVID-19	162.46
General	COVID-19	182.74
	no-COVID-19	144.32

**Table 8 ijerph-19-09886-t008:** Mann–Whitney U Test with a no COVID-19–COVID-19 grouping item (post-test).

Mann–Whitney U Test							
	DS_1.	DS_2.	DS_3.	DS_4.	DS_5.	DS_6.	General
Mann–Whitney U Test	5687.000	6475.500	6249.500	6155.500	6379.500	5250.000	6331.000
W de Wilcoxon	10,257.000	15,938.500	10,810.500	15,610.500	15,834.500	14,702.000	15,784.000
Z	−1.625	−0.044	−0.514	−0.703	−0.252	−2.509	−0.351
Sig. Asymptotic (bilateral)	0.005	0.001	0.005	0.010	0.000	0.001	0.000

Grouping item: no COVID-19/COVID-19.

**Table 9 ijerph-19-09886-t009:** Post-test average ranges by DS factors.

DS Factors	COVID-19/Non-COVID-19	Average Range
DS_1. Information DS	COVID-19	142.42
	no-COVID-19	107.97
DS_2. Communication DS	COVID-19	136.34
	no-COVID-19	116.73
DS_3. Collaboration DS	COVID-19	148.38
	no-COVID-19	113.79
DS_4 Critical thinking DS	COVID-19	173.89
	no-COVID-19	143.95
DS_5. Creative DS	COVID-19	145.58
	no-COVID-19	117.83
DS_6. Problem-solving DS	COVID-19	137.31
	no-COVID-19	129.75
General	COVID-19	145.21
	no-COVID-19	118.36

**Table 10 ijerph-19-09886-t010:** Rho Spearman Correlation for usage time of ICT and the dimensions of DS in both groups A and B.

Rho Spearman Correlation for Usage Time of ICT and the Dimensions of DS										
				**DS_1.**	**DS_2.**	**DS_3.**	**DS_4.**	**DS_5.**	**DS_5.**	**General**
Spearman’s Rho Group A	How long have you been utilising technology as a tool in your classroom?	Correlation coefficient	1.000	0.040	0.084	0.047	0.013	0.054	0.034	0.058
		Sig. (bilateral)		0.526	0.182	0.469	0.855	0.372	0.565	0.355
		N	232	232	232	232	232	232	232	232
				**DS_1.**	**DS_2.**	**DS_3.**	**DS_4.**	**DS_5.**	**DS_5.**	**General**
Spearman’s Rho Group B	How long have you been utilising technology as a tool in your classroom?	Correlation coefficient	1.000	0.224 **	0.198 **	0.301 **	0.223 **	0.280 **	0.351 **	0.319 **
		Sig. (bilateral)		0.000	0.001	0.001	0.000	0.001	0.000	0.000
		N	237	237	237	237	237	237	237	237

** The correlation is significant at the 0.01 level (bilateral).

## Data Availability

Not applicable.

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
