# Peer review of "Before and Amid COVID-19 Pandemic, Self-Perception of Digital Skills in Saudi Arabia Higher Education: A Longitudinal Study"

_ijerph, 2022, doi:10.3390/ijerph19169886_

Round 1
Reviewer 1 Report
The article presents a very important issue of digital competences. The structure of the article is correct.
Remarks:
- the abstract should clearly define the purpose of the research,
- in wnisokach, it is worth emphasizing the implementation of the adopted goal;
- in addition, highlight the limitations of the research in the summary and outline the further perspective of the research (this information was not found in any part of the article),
- the value of the article would be greater if the Authors clearly presented in the Student Digital Skills part, e.g. in the form of a table, what digital skills are, including information and IT skills, and justified the presented division.
Author Response
Dear reviewer
Thank you for giving us the opportunity to submit a revised draft of our manuscript entitled “Before and Amid COVID-19 Pandemic, Self-Perception of Digital Skills in Saudi Arabia Higher Education: A longitudinal Study”. We appreciate the time and effort you have dedicated to providing us with your valuable feedback on our manuscript. We have colored the changes within the manuscript in red. attached is a point-by-point response to your comments and concerns.

Reviewer 2 Report
Interesting paper.
The authors should improve the exposition and the flow mostly in the introduction. The abstract needs to be revised as well.
Results are clear. Conclusion could be improved with few additional explanations, mostly on the connection of the results with previous research works.
Please refer to red comments in the text.

Author Response
Thank you for giving us the opportunity to submit a revised draft of our manuscript entitled “Before and Amid COVID-19 Pandemic, Self-Perception of Digital Skills in Saudi Arabia Higher Education: A longitudinal Study”. We appreciate the time and effort you have dedicated to providing us with your valuable feedback on our manuscript. We have colored the changes within the manuscript in red. attached is a point-by-point response to your comments and concerns.

Reviewer 3 Report
The manuscript addresses a very interesting topic at the international level that can help to improve the digital skills of students. Under the title "Before and Amid COVID-19 Pandemic, Self-Perception of Digital Skills in Saudi Arabia Higher Education: A longitudinal Study", the authors offer this Article from the Special Issue “Higher Education and Sustainable Development Goals (SDG) in Good-Health and Well-Being, Education and Social Inclusion". I find the topic of the manuscript as well as review of the up-to-date literature provided by the authors are appropriate.
The overall assessment of the paper is positive. The authors have pointed out the importance of the self-perception of digital skills of students in higher education before and amid the COVID-19 pandemic with a longitudinal study.
The authors address the topic of the study in a clear way and rely on a review of the scientific literature that is comprehensive and current.
METHOD. The research design is correct and the data used seem useful for the objectives set by the author of the text. Regarding Research Context and Participants:
• The population is unknown. Some information is necessary.
• Sample of the Study. It states that the sample is 469 students: 232 students from the 130 2019–2020 academic year (group A) and 237 students from the 2020–2021 academic year 131 (group B), but it does not provide data about their universities. I suggest a table with this information.
• The sample is important, but a larger sample is necessary to carry out a study differentiated by gender. A study differentiated by gender would have been interesting. This point would have provided value to the study.
• Some information about data collection process is necessary.
Overall it is a good piece of work and I hope my suggestions will help you.
Author Response

(The authors gave the same response as above.)
